# Different Methods to Modify the Hydrophilicity of Titanium Implants with Biomimetic Surface Topography to Induce Variable Responses in Bone Marrow Stromal Cells

**DOI:** 10.3390/biomimetics9040227

**Published:** 2024-04-10

**Authors:** Thomas W. Jacobs, Jonathan T. Dillon, David J. Cohen, Barbara D. Boyan, Zvi Schwartz

**Affiliations:** 1Department of Pharmaceutics, Virginia Commonwealth University, Richmond, VA 23298, USA; twjacobs@vcu.edu; 2Department of Biomedical Engineering, College of Engineering, Virginia Commonwealth University, 601 West Main Street, Richmond, VA 23284, USA; dillonj4@vcu.edu (J.T.D.); djcohen@vcu.edu (D.J.C.); bboyan@vcu.edu (B.D.B.); 3Wallace H. Coulter Department of Biomedical Engineering, Georgia Institute of Technology, Atlanta, GA 30332, USA; 4Department of Periodontics, The University of Texas Health Science Center at San Antonio, San Antonio, TX 78229, USA

**Keywords:** plasma treatment, wettability, titanium, grit-blasted/acid-etched implant surfaces, osteogenesis

## Abstract

The osteoblastic differentiation of bone marrow stromal cells (bMSCs), critical to the osseointegration of titanium implants, is enhanced on titanium surfaces with biomimetic topography, and this is further enhanced when the surfaces are hydrophilic. This is a result of changing the surface free energy to change protein adsorption, improving cell attachment and differentiation, and improving bone-to-implant contact in patients. In this study, we examined different methods of plasma treatment, a well-accepted method of increasing hydrophilicity, and evaluated changes in surface properties as well as the response of bMSCs in vitro. Commercially pure Ti and titanium–aluminum–vanadium (Ti6Al4V) disks were sand-blasted and acid-etched to impart microscale and nanoscale roughness, followed by treatment with various post-processing surface modification methods, including ultraviolet light (UV), dielectric barrier discharge (DBD)-generated plasma, and plasma treatment under an argon or oxygen atmosphere. Surface wettability was based on a sessile water drop measurement of contact angle; the elemental composition was analyzed using XPS, and changes in topography were characterized using scanning electron microscopy (SEM) and confocal imaging. The cell response was evaluated using bMSCs; outcome measures included the production of osteogenic markers, paracrine signaling factors, and immunomodulatory cytokines. All plasma treatments were effective in inducing superhydrophilic surfaces. Small but significant increases in surface roughness were observed following UV, DBD and argon plasma treatment. No other modifications to surface topography were noted. However, the relative composition of Ti, O, and C varied with the treatment method. The cell response to these hydrophilic surfaces depended on the plasma treatment method used. DBD plasma treatment significantly enhanced the osteogenic response of the bMSCs. In contrast, the bMSC response to argon plasma-treated surfaces was varied, with an increase in OPG production but a decrease in OCN production. These results indicate that post-packaging methods that increased hydrophilicity as measured by contact angle did not change the surface free energy in the same way, and accordingly, cells responded differently. Wettability and surface chemistry alone are not enough to declare whether an implant has an improved osteogenic effect and do not fully explain how surface free energy affects cell response.

## 1. Introduction

Titanium-based dental implants are the best and most prominently used method for replacing teeth due to their corrosion-resistant biocompatible surfaces and functional mechanical properties, resulting in excellent success rates and esthetics [1]. Implant success relies on their integration with the surrounding bone. However, dental implants are often placed in older patients or those with comorbidities, such as diabetes and osteoporosis, which reduce retention rates and contribute to reduced long-term effectiveness [2]. Thus, it is critical to design technologies that positively affect implant osseointegration to provide the highest chance of retention in these compromised patient populations.

This process of implant osseointegration is governed by a variety of factors, including the bone quality of the patient, the implant material and design, and surface properties such as roughness and surface free energy. During the process of implant osseointegration, bone marrow stromal cells (bMSCs) are among the first types of cells recruited to the site of implantation. These cells are responsible for modulating the key signaling pathways that regulate angiogenesis, osteogenesis, and local immune response, resulting in the integration of an implant with native bone [3].

Chief among these processes is the attachment and differentiation of bMSCs into osteoblasts, the cells responsible for new bone formation. In vitro studies show that surface modifications to titanium implants that impart microscale and nanoscale roughened topography using sand-blasting and acid-etching can induce the differentiation of bMSCs into osteoblasts without the need for osteogenic media supplementation [4,5,6,7]. These implants have a biomimetic surface topography that resembles the surface of bone following its resorption by osteoclasts. BMSCs and osteoprogenitor cells migrate onto the osteoclast-conditioned bone surface and synthesize and mineralize the bone extracellular matrix. In vitro and in vivo studies indicate that a similar series of events occurs on Ti substrates with the biomimetic topography [8,9].

Manufacturing the biomimetic surfaces in a nitrogen environment, or through the use of post-packaging modifications such as UV light or plasma treatment, results in implant surfaces that are superhydrophilic and further improve cellular response in vitro and osseointegration in vivo [10,11,12,13,14,15]. In recent years, the use of plasma treatment as a post-packaging modification to enhance the osseointegration of implants has been a particular area of research due to the development of benchtop plasma devices that have realistic clinical translatability and can potentially improve patient outcomes with little risk [16].

The wettability of an implant surface is often determined using sessile water drop contact angle measurements and is considered to be an indicator of surface free energy. Surface free energy, which is a measure of unsatisfied bond energy on a material’s surface, plays a critical role in how biological fluids react when they first come into contact with the implant. Atoms on the surface have fewer neighbors than those in the bulk material, giving rise to unsatisfied bond energy known as “dangling bonds” [17]. These bonds can either be primary (covalent, ionic, or metallic) or secondary (van der Waals forces) and contribute in different ways to the behavior of the surface. A surface free energy comprised largely of bonds resulting from van der Waals forces will behave in a more non-polar nature and increase hydrophobicity, while stronger covalent or ionic bond forces will exhibit more Lewis acid and base properties that increase the hydrophilicity of the implant. Thus, surface free energy is a measure of the energy on the surface of the implant based on the type and number of these dangling bonds. Protein adsorption, cell attachment, and water interactions can all be influenced by this unsatisfied bond energy [18]. Hydrophilic surfaces are reported to enhance cell attachment [19], increase the production of osteogenic factors and osteoblast markers [20,21], and improve bone-to-implant contact [22,23] compared to hydrophobic surfaces—making quantification of surface free energy a key component in implant surface characterization.

Plasma treatment has proven to be an effective way to achieve super-hydrophilic surfaces on Ti implants [10,24]. However, plasma treatment is not without its own set of challenges. First, there is evidence that wettability is not a truly representative measure of surface free energy. Water, used in these sessile drop measurements of contact angle, is not a perfect representation of the biological fluids that come into contact with the implant. Proteins and other ions present in the body can create strong acid–base interactions between the surface and the fluid based on the actual surface free energy, resulting in differences from the measured wettability [18]. Furthermore, plasma treatment is only temporary. During plasma cleaning, hydrocarbons on the surface are removed. These hydrocarbons are predominantly non-polar in nature and have very low surface free energies that contribute to the natural hydrophobic nature of the implant [10,18]. After removal through plasma treatment, the implant adopts the desired hydrophilic state, as measured by wettability, but the surfaces eventually return to their normal hydrophobic state [25]. This is a result of atmospheric hydrocarbons rapidly adsorbing to the surface when in contact with the atmosphere, creating the hydrophobic state once again and limiting the desired cell response. To remedy this, medical device companies have begun to design and produce benchtop plasma cleaners for the operating room capable of treating implants just prior to surgical placement. This limits exposure to hydrocarbons that would return surfaces to their normal hydrophobic state.

In this study, we compared the effects of different post-packaging modifications, including three plasma treatment technologies and a benchtop UV light device, on Ti surface wettability and chemistry and examined the response of bMSCs to the resulting surface changes. Technologies included a benchtop UV-based cleaning device that exposes implant surfaces to UV light for 10 s, sterilizing the surface and improving hydrophilicity. We also tested a benchtop-based plasma cleaner that uses dielectric barrier discharge (DBD) under moderate vacuum conditions (5–10 Torr) maintained for 60 s. DBD takes advantage of high-frequency radio waves to initiate plasma discharge. There is also evidence that DBD treatment can affect surface crystallinity dependent on voltage, frequency, and time of exposure while avoiding the addition of metallic ions that could inhibit hydrophilicity [26]. Finally, we tested a device that generates plasma using either argon or oxygen-enriched gas and treats the surface for variable durations. Plasma treatment was used to treat sand-blasted/acid-etched Ti and Ti6Al4V surfaces, including Ti surfaces that were manufactured in a nitrogen-rich environment and packaged in saline (modSLA) to impart hydrophilic properties and prevent atmospheric hydrocarbon deposition. All surfaces used in the study had a biomimetic surface topography with microscale and nanoscale features typical of osteoclast resorption pits on bone.

## 2. Materials and Methods

### 2.1. Surface Manufacturing

Titanium–aluminum–vanadium (Ti6Al4V) substrates were produced as previously described [10]. Briefly, grade 4 Ti6Al4V rods were milled into 10.5 mm by 5.25 mm rectangular surfaces and treated by grit-blasting and acid-etching (GB + AE) using proprietary technology (AB Dental, Ashdod, Israel). Grit-blasting consisted of using calcium phosphate particles followed by degreasing and bathing in HNO_3_ for 5 min. Acid-etching was accomplished using a series of proprietary acid washes with H_2_SO_4_ and HCl. Following acid washes, surfaces were rinsed 3 times in ultrapure distilled H_2_O for 10 min. Surfaces were then blotted, air-dried, and packaged. Surfaces were sterilized using gamma radiation.

Ti disks were prepared as described previously and sterilized with 25 kGy gamma irradiation prior to use [5]. In brief, 15 mm diameter disks were punched from 1 mm thick sheets of grade 2 Ti (Institut Straumann AG, Basel, Switzerland), degreased in acetone, and processed for 30 s in 55 °C 2% ammonium fluoride/2% hydrofluoric acid/10% nitric acid solution to produce pretreatment Ti disks (PT). SLA substrates were prepared by subjecting PT surfaces to sand-blasting (250–500 μm corundum) and acid-etching (HCl/H_2_SO_4_). Disks were cleaned in HNO_3,_ rinsed in deionized water, air-dried, and packed in aluminum foil. ModSLA surfaces were produced in the same manner of sand-blasting and acid-etching as SLA surfaces, except subsequent steps took place under nitrogen gas to prevent exposure to air. The modSLA surfaces were rinsed and stored in 0.9% NaCl solution.

### 2.2. UV Light Treatment

Ti6Al4V substrates were placed on a magnetic mount and attached to the retractable machine mount. Upon initiation of treatment the mount withdrew into the device and a vacuum was created. UV treatment was initiated using a radio frequency of 13.56 MHz with maximum power output of 0.001795 mW, and UV exposure was set to 172 nm wavelength. Treatment duration was 10 s.

### 2.3. Dielectric Barrier Discharge Plasma Cleaning

Ti6Al4V substrates were attached to a magnetic, electrically grounded supporting mount and placed inside the machine. The column wall consisted of transparent polypropylene, which served as a dielectric barrier layer and automatically descended upon starting treatment. Plasma initiation was achieved as previously described [10]. In brief, a sinusoidal electric power with a frequency of 100 kHz and voltage of 3 kV was applied to an external electrode to generate a dielectric barrier discharge on the surface that was maintained for 60 s. A hollow needle accessed the interior of the column through the silicon rubber sealing cover. This needle was connected to a vacuum pump to regulate pressure inside the column at 5–10 Torr for the duration of the treatment.

### 2.4. Argon and Oxygen Plasma

Substrates were placed on an aluminum mesh mount and set into a Solarus plasma cleaner (Solarus Model 950, Gatan Inc., Pleasanton, CA, USA). For oxygen plasma treatment, gas flow was set at 35.0 sccm O_2_ at 50 W for 10 min. During argon treatment, argon gas flow was set to 11.5 sccm Ar at 50 W for 10 min. Two different studies were conducted. In the first study, Ti6Al4V surfaces were treated with argon plasma. In the second study, Ti SLA and modSLA surfaces were treated with either argon plasma or oxygen plasma. This experimental design enabled us to compare the effects of argon on Ti vs. Ti6Al4V. In addition, we compared the effects of plasma treatment on a surface that was already hydrophilic (modSLA) to one that was hydrophobic (SLA).

### 2.5. Surface Characterization

#### 2.5.1. Scanning Electron Microscopy

Surface topography and morphology were qualitatively visualized using scanning electron microscopy (SEM; Hitachi SU-70, Tokyo, Japan). Substrates were placed on SEM imaging mounts using carbon tape and imaged with 56 μA ion current, 5 kV accelerating voltage at a 5 mm working distance. Surfaces were imaged at 6 different locations on two separate surfaces at multiple resolutions, and representative images are presented in the Appendix A.

#### 2.5.2. Contact Angle Analysis

Contact angle analysis was performed using water in a sessile drop test measured using a goniometer (CAM 250, Ramé-Hart Instrument Co. Succasuna, NJ, USA). Contact angles were measured in 6 different locations on two different surfaces (*n* = 12); surfaces were dried with nitrogen gas between measurements. A 3 μL droplet of distilled water was used for each measurement.

#### 2.5.3. Roughness Analysis

Optical profilometry to measure surface topography was performed using a confocal microscope (Zeiss LSM 710, Carl-Zeiss AG, Oberkochen, Germany), employing a main beam splitter set to T80/R20 with reflectance. Z-stacks were taken at 1.00 μm intervals using a high pass filter with a cut-off at 20 μm. Measurements were made at 6 different locations on two different surfaces (*n* = 12).

#### 2.5.4. Chemical Analysis

Element composition was analyzed using X-ray photoelectron spectroscopy (XPS) (PHI VersaProbe III Scanning XPS, Physical Electronics Inc., Chanhassen, MN, USA). Samples were secured to the instrument mount using copper clips. The mount had been cleaned via sonication in ethanol solution prior to use. Analysis was performed using a 50-Watt, 15 kV X-ray gun with a spot size of 200 μM, 20 ms dwelling time, and 1 eV step size. Analysis was performed on two samples per treatment group at six different positions on the surface (*n* = 12). Representative survey and region scan spectra are shown in the Appendix A.

### 2.6. Cell Culture

Cell culture on pure Ti and Ti alloy surfaces was performed as described previously [27]. In brief, human male bMSCs (Ossium Health, San Francisco, CA, USA) were cultured in MSC growth medium (GM) comprised of αMEM with 4 μM L-glutamine and 10% fetal bovine serum (FBS) at 37 °C in 5% CO_2_ and 100% humidity. At 80% confluence in T75 flasks, cells were trypsinized and seeded onto surfaces 9500 cells in 0.5 mL per well.

For rectangular Ti6Al4V substrates, two surfaces of the same group were placed side-by-side per well in 24-well plates. Cells were allowed to attach to the surfaces for 24 h before being carefully removed and placed into new 24-well plates to ensure that only cells attached to the surfaces were assessed. Circular 15 mm diameter SLA and modSLA Ti disks were placed individually into each well.

Cells were cultured on tissue culture polystyrene (TCPS) as an optical control for all experiments. Six wells per variable (TCPS, untreated surface and treated surface) were plated for each experiment. GM was changed every 48 h thereafter. On day 7, cells were incubated for 24 h in fresh GM before harvesting. Conditioned media were collected and immediately stored at −80 °C. Cell layer lysates were rinsed twice with 1 mL 1X PBS and lysed in 0.5 mL 0.5% Triton X-100 and immediately stored at −80 °C for biological assays.

### 2.7. Cellular Response

Cell response was evaluated as previously described [27]. In brief, cell layers were suspended in 0.5% Triton X-100 and lysed by ultrasonication at 40 V for 10 s/well (VCX 130; Vibra-Cell, Newtown, CT, USA). Total DNA content was measured using the QuantiFluor dsDNA system (Promega, Madison, WI, USA). Enzyme-linked immunosorbent assays (ELISAs) were performed to determine levels of osteogenic markers, immunomodulatory cytokines, and paracrine signaling factors in the harvested conditioned media. Osteocalcin (DY1419-05, OCN), osteopontin (DY1433, OPN), osteoprotegerin (DY805, OPG), vascular endothelial growth factor 165 (DY293B-05, VEGF), interleukin-6 (DY206, IL-6), and interleukin-10 (DY217B, IL-10) were quantified according to the manufacturer’s protocols (R&D Systems, Inc., Minneapolis, MN, USA). Production of proteins was normalized to total DNA content.

### 2.8. Statistical Analysis

Data collected are means ± standard error of six independent cultures per variable. All experiments were repeated to ensure validity of analysis, with results of individual experiments shown. Statistical analysis among groups was evaluated by Student’s unpaired *t*-test or one-way analysis of variance (ANOVA), and multiple comparisons between groups were conducted with a two-tailed Tukey post-test. A *p*-value less than 0.05 was considered statistically significant. All statistical analysis was performed using the GraphPad Prism v10.02 software.

## 3. Results

### 3.1. Effect of UV–Plasma Treatment

#### 3.1.1. Surface Properties

Prior to UV–plasma cleaning, Ti6Al4V surfaces were found to be hydrophobic with a contact angle of 82 degrees (mean, *n* = 12) (Figure 1A,C). After treatment with the UV–plasma, the surfaces were significantly more hydrophilic, with a hardly discernible water drop (Figure 1B), and the contact angle measurement was significantly reduced at just 10 degrees (Figure 1C). Arithmetic mean height, a measure of the surface roughness of the sample, was significantly increased following UV–plasma treatment (Figure 1D), but the peak-to-valley distance was unchanged (Figure 1E). Surface chemistry was also changed in response to UV–plasma cleaning. Following treatment, levels of oxygen and titanium on the surface were increased compared to the untreated control, while levels of carbon were decreased (Figure 1F). This loss of hydrocarbons can, in part, account for the hydrophilicity measured by the contact angle of the water drop. Results confirmed that treatment imparted increased hydrophilicity while maintaining similar roughness of the surface. Morphologically, surfaces were visually the same after treatment. Scanning electron microscopy showed that nanostructured ridgelines and pits were remarkably similar before and after UV–plasma treatment (Appendix A).

#### 3.1.2. BMSC Response

BMSCs were cultured on TCPS, Ti6Al4V surfaces, and UV–plasma-treated Ti6Al4V surfaces. In comparison to cells cultured on TCPS, the total DNA content was significantly reduced on both treated and untreated SLA surfaces, with no difference between treated and untreated substrates (Figure 2A). Osteoblast markers OCN and OPN were significantly increased in cells cultured on the UV–plasma treatment surfaces compared to TCPS while OPG was decreased. OPN and OPG production on surfaces without UV treatment were not significantly different from TCPS. However, these osteogenic markers showed no statistical difference whether surfaces were treated with UV–plasma or not, despite being significantly more hydrophilic after treatment (Figure 2B–D). Similarly, VEGF, a paracrine signaling factor for angiogenesis, and the pro-inflammatory cytokine IL-6 were decreased compared to the TCPS control on SLA surfaces, with no difference as a function of treatment (Figure 2E,F). The anti-inflammatory cytokine IL-10 was increased on Ti6Al4V surfaces +/− treatment compared to TCPS (Figure 2G).

### 3.2. Effect of DBD Plasma

#### 3.2.1. Surface Properties

Prior to plasma cleaning, Ti6Al4V surfaces were highly hydrophobic with obvious beading of the water droplet and a contact angle measurement of 115 degrees (Figure 3A,C). After DBD-plasma treatment, surfaces were superhydrophilic with no obvious droplet beading and a contact angle measurement of 5 degrees (Figure 3B,C). Again, treatment seemed to change surface roughness slightly, with a measured arithmetic mean deviation that was significantly different after treatment (Figure 3D). The measured peak-to-valley distance of the surfaces was unchanged (Figure 3E). Treatment increased the oxygen content of the surface while relative levels of carbon and Ti were decreased (Figure 3F). There were no visible morphological changes in micro- or nano-structure architecture (Appendix A).

#### 3.2.2. Cell Response

Total DNA content was reduced by 30% following plasma treatment (Figure 4A). The OCN, OPN, and OPG contents of the conditioned media were elevated in the cultures grown on DBD-treated Ti6Al4V (Figure 4B–D). In contrast, VEGF production was not affected by plasma cleaning, nor was the pro-inflammatory cytokine IL-6 (Figure 4E,F). However, the anti-inflammatory cytokine IL-10 production was significantly increased in cells cultured on the plasma-cleaned surfaces (Figure 4G).

### 3.3. Effect of Argon Plasma

#### 3.3.1. Ti6Al4V Surface Properties

Untreated Ti6Al4V surfaces showed characteristic hydrophobicity with obvious beading of the water droplet before treatment (Figure 5A), with a contact angle of 60 degrees (Figure 5C). Following argon plasma treatment, surfaces were markedly more hydrophilic with very little beading of the water droplet and contact angle below 10 degrees (Figure 5B,C). The same increase in surface roughness measured by arithmetic mean deviation that was seen on surfaces treated with UV–plasma cleaning was also measured after argon treatment, while the average peak-to-valley distance was unchanged (Figure 5D,E). SEM imaging of the surfaces before and after argon treatment showed no obvious visual changes to the micro- and nano-structures (Appendix A). After argon treatment, there was a decrease in the relative content of oxygen and titanium but an increase in carbon content on the surface (Figure 5F).

#### 3.3.2. Cell Response to Ti6Al4V Surfaces

There was no change in total DNA content as a result of the argon treatment of the Ti6Al4V surfaces (Figure 6A). OCN was decreased (Figure 6B), OPN was unchanged (Figure 6C), and OPG was increased (Figure 6C) in cultures grown on the argon-treated surfaces. No changes were observed in the production of VEGF, IL-6, or IL-10 (Figure 6E–G).

### 3.4. Effect of Argon and Oxygen Plasmas on Ti Surfaces

#### 3.4.1. Surface Properties

Treatment of hydrophobic SLA surfaces with argon or oxygen plasmas reduced contact angles to less than 5 degrees (Figure 7A). Untreated modSLA surfaces were hydrophilic, and this was conserved after plasma treatment (Figure 7B). Neither argon nor oxygen treatment altered the RSa or RSz of SLA or modSLA (Figure 7C–F). Surface chemistry analysis showed relatively increased levels of oxygen and titanium and decreased carbon content following argon and oxygen plasma treatment compared to the untreated SLA control (Figure 7G).

#### 3.4.2. Cell Responses to SLA-O_2_ and SLA-AR

Although there were no differences in surface properties, the response of bMSCs varied with the type of surface (SLA vs. modSLA) and with plasma treatment (argon vs. oxygen) (Figure 8). DNA content was lowest in cultures grown on modSLA compared to SLA, SLA treated with argon (SLA-AR), and SLA treated with oxygen (SLA-O_2_) (Figure 8A). In contrast, the production of OCN, OPN, and OPG was highest in these cultures (Figure 8B–D). The production of IL-6 was reduced in cultures grown on SLA treated with argon or oxygen but not to the same extent as in cultures grown on modSLA (Figure 8E). IL-10 production was increased in the treated SLA cultures to levels comparable to cultures grown on modSLA (Figure 8F). Total DNA content was significantly decreased on modSLA surfaces compared to the SLA control. Surfaces treated with argon or oxygen-based plasma treatment for 10 min did not have a significant effect on total DNA content (Figure 8A). Osteogenic markers OCN, OPN, and OPG were all increased for the modSLA group compared to the SLA control. Plasma cleaning did not have a significant effect on the production of osteogenic markers (Figure 8B–D). Interestingly, IL-6 was decreased on the hydrophilic surfaces compared to the SLA control, and the modSLA group had the lowest level of IL-6 production (Figure 8E). IL-10 production was significantly increased on the hydrophilic surfaces compared to the SLA group (Figure 8F).

#### 3.4.3. Cell Response to Plasma-Treated modSLA

The DNA content in bMSC cultures grown on modSLA was reduced compared to cultures on SLA; the argon treatment of the SLA and modSLA substrates did not alter this difference (Figure 9A). However, osteogenic factor production by cells on modSLA was impacted by plasma treatment. Untreated modSLA substrates supported increased levels of OCN, OPN, and OPG compared to the SLA surfaces. Argon plasma treatment of the modSLA surfaces decreased OCN and OPN production compared to the untreated modSLA surfaces, while there was no observed effect on OPG production. Argon plasma treatment of the SLA surface did not affect levels of osteogenic markers compared to the untreated SLA surface (Figure 9B–D). Similarly, the analysis of immunomodulatory cytokines showed decreases in IL-6 production and increases in IL-10 production in the modSLA groups compared to the SLA surfaces. There were no observed differences in production for argon plasma-cleaned surfaces versus their uncleaned counterparts (Figure 9E,F).

## 4. Discussion

Our results demonstrate that although each post-packaging treatment method produced hydrophilic surfaces, not all the methods examined improved osteogenic differentiation of bMSCs. An analysis of the surface properties before and after treatment demonstrated that small but significant differences possibly contributed to the variability in cell response. Overall, all the plasma treatment modalities improved the hydrophilicity of surfaces without depositing additional metal ions. In general, the microscale and nanoscale topography were retained, although there were some plasma-related changes in physical properties, as discussed below. The most striking plasma-related changes were due to alterations in surface chemistry.

Plasma treatment resulted in relative increases in surface Ti and oxygen and reductions in carbon due to the removal of contaminating hydrocarbons. These hydrocarbons are predominantly non-polar in nature and have very low surface free energies that contribute to the natural hydrophobic nature of the implant [10,18]. After removal through plasma treatment, the implant surface adopts the desired hydrophilic state, as measured by wettability. Ti6Al4V substrates treated with argon plasma exhibited decreased oxygen and an increase in relative carbon content, but the surface was still observed to be significantly more hydrophilic.

Plasma treatment eliminates adsorbed hydrocarbons by moving ionized gas particles across the surface within the sample chamber, creating impact forces and microcombustions that convert elements to gas that are then removed from the chamber. Despite this, plasma treatment to increase surface hydrophilicity is not always correlated to a reduction in carbon content, as shown by the argon plasma treatment and in previous studies [28]. Not all benchtop plasma devices are designed to remove adsorbed hydrocarbons. If the device is not equipped with a vacuum to remove molecules as they are lifted from the surface by the plasma, they will redeposit onto the surface after treatment is finished. The hydrophilicity of the surface is improved by the treatment, although carbon content is not reduced. Thus, surface carbon content is not a clear indicator of hydrophilicity.

An increase in the relative content of titanium and oxygen on the surface was found to be beneficial for osteoblast maturation, indicating that measured cell response should be enhanced by plasma treatment [29]. Our results show that the effects of surface chemistry on bMSC response are more nuanced. The change or lack thereof in carbon content is not an indicator of cell response. In this study, we showed multiple plasma treatments (UV, DBD and extended oxygen) reduced surface carbon content, but had varied cellular responses; a treatment that increased carbon content (argon treatment) and produced a negative cell response, and in prior studies, showed a surface with enhanced hydrophilicity and no change in carbon content, enhanced cell and in vivo response [10,28]. We can conclude that while carbon content is involved in wettability and the surface free energy of an implant, it is not the only factor in determining cell response or hydrophilicity.

Plasma cleaning slightly modified surface roughness, increasing arithmetic mean deviation in surfaces treated by UV–plasma, argon, and DBD. The peak-to-valley distance was unaffected by treatment. Surface roughness can play a critical role in the surface free energy of an implant and the measured wettability, though the reason why is still unclear. Some researchers theorize that air molecules become trapped in the micro- and nanoscale architecture, creating an inhomogeneous and hydrophobic surface/air–liquid interface during testing [30]. These variations in surface topography contribute to the complexity of evaluating surface free energy, further distinguishing wettability measurements from representing the true surface free energy of the implant and how cells will behave when in contact [18].

Cell responses to plasma treatment varied. Previous research correlated an increase in hydrophilicity with an enhanced osteogenic response both in vivo and in vitro [13,14,28]. These studies were performed with a very specific set of conditions, using Ti6Al4V surfaces and implants. In the current study, surfaces were characterized in the same fashion as previously described, and the in vitro biological response was evaluated in the same manner [10]. Based on this, we can conclude that although all surfaces had increased wettability following treatment, only the methods that showed a positive osteogenic response in the cells would correlate with an improved osteogenic response in animals [28].

UV, argon, and oxygen plasma treatments all altered cell response to the substrate surface. Argon plasma decreased OCN production on Ti6Al4V surfaces and decreased the inflammatory cytokine production of SLA surfaces compared to the untreated control cultures. Similarly, oxygen plasma treatment increased IL-10 production and decreased IL-6 production on SLA surfaces. Neither plasma treatment improved the osteogenic response to SLA surfaces compared to modSLA surfaces. In contrast, DBD plasma under vacuum conditions robustly enhanced the osteogenic response to the surfaces. This reinforces the idea that using a water drop contact angle to assess wettability is not sufficient to determine if a surface will be osteogenic and is not a complete measure of surface free energy. While the growth media used in cell culture is largely comprised of water, it is possible that ions and other additives, including the fetal bovine serum necessary for cell growth, adsorb to the surface differently as a function of altered surface free energy of the substrate and, thus, the way cells behave. This is similar to the idea that proteins and other ions present in the fluid that comes into contact with an implant during its insertion can create strong acid–base interactions based on the actual surface free energy. Thus, simply measuring wettability based on contact angle measurements of water droplets is not sufficient in declaring a hydrophilic implant will elicit an improved osteogenic response and osseointegration in vivo.

Further examination of in vitro cell response is necessary to fully characterize the effects of surface plasma treatments. Gentleman et al. [18] previously drew attention to how wettability lacks precision in predicting cell–biomaterial interactions, particularly in the case of surfaces of varying topographies, calling for improvements to be made in surface characterization experiments that decouple surface free energy from surface roughness. Research into the surface free energy of biomaterials progressed to better elucidate surface properties based on wettability by considering factors such as contact angle hysteresis by vibrating the surface during testing and measuring the contact angle of fluids with varying surface tensions, densities, and viscosities such as in the Owens–Wendt method [31,32,33]. However, a clear link between these enhanced wettability measurements, plasma treatment of titanium surfaces, and the resulting osteogenic response has not been studied and warrants further exploration.

Our study design allowed us to compare the osteogenic response to biomimetic Ti surfaces that were processed under conditions to retain hydrophilicity to the osteogenic response on Ti surfaces that were made hydrophilic via plasma treatment. Previous work shows that the modSLA implant surface, which is prepared in a nitrogen environment and stored in saline conditions to prevent hydrocarbon deposition from the environment, supports improved osteoblast differentiation compared to SLA in vitro and enhanced osseointegration in vivo [34]. In the present study, we found that plasma treatment of modSLA further increased the pro-osteogenic response to the substrate for some but not all parameters. The argon plasma treatment of modSLA surfaces reduced the production of OPN compared to untreated modSLA. The reasons for this are not clear. The processing of modSLA introduces nano-texture to the SLA surface, which was retained following argon plasma treatment. If modSLA surfaces are allowed to age under normal atmospheric conditions, the nano-texture is retained, but the surface becomes hydrophobic and cells respond to it as if it were SLA [34]. This suggests that the argon plasma treatment introduced a further modification to the surface that affected the production of OPN via signaling pathways independent of OCN, potentially via the adsorption of a specific set of atmospheric hydrocarbons.

Clinically, special care should be taken when using plasma cleaners to enhance the osteogenic potential of implants. As shown in this study, despite each plasma treatment creating the desired hydrophilic state of the surface as measured by wettability, not all in vitro responses to the surface were positive, indicating that wettability is not a clear measurement of implant surface free energy or fully correlated to cell response. More analysis must be performed on implant surfaces and plasma cleaners than simply declaring a hydrophilic surface to have an increased osteogenic effect based on wettability testing. There is evidence that protein adsorption measured by the adhesion of bovine serum albumin (BSA) can be linked to enhanced osteogenic differentiation and the osseointegration of titanium implants [35]. Future studies are planned to examine the relationship between protein adsorption as a measurement of surface free energy and cell response on plasma-treated surfaces, as well as using different simulated body fluids to test surface wettability correlated to osteogenic response. There is a need for improved methods of measuring surface free energy beyond wettability to correlate osteogenic properties; moreover, surfaces require in vitro examination when evaluating plasma treatments, and care should be taken when considering the veracity of plasma cleaners for use in the clinic.

## 5. Conclusions

In this study, we compared the effects of four different post-packaging surface modifications on the wettability of biomimetic Ti surfaces as measured by the water contact angle and examined the effect of this induced hydrophilicity on the differentiation of bMSCs. All treatments were effective in inducing superhydrophilic surfaces with minimal changes to surface topography. Notably, treatments tended to reduce carbon content on the surface with the exception of argon gas plasma treatment, which increased carbon content by a small but significant amount compared to the untreated control surface. The resulting cell response after this treatment showed decreased OCN production, increased OPG production, and no differences among the other factors measured. Based on these results, the treatment did not have a positive effect in inducing osteoblast differentiation of bMSCs. This is in contrast to the other treatments that enhanced the osteogenic responses of the surfaces. Notably, all osteogenic markers were increased, and total DNA content decreased following DBD plasma treatment, indicating increased osteoblast differentiation. The differences in cell response presented here depended on surface treatment despite inducing superhydrophilicity on all surfaces, indicating that measuring hydrophilicity using water contact angle measurements is not a reliable indicator of the enhancement of the osteogenic response of an implant or surface. Thus, not all treatments affected the surface free energy in the same fashion, and accordingly, cells responded differently. Clinically, the induced surface hydrophilicity of an implant will not always enhance osseointegration.

## Figures and Tables

**Figure 1 biomimetics-09-00227-f001:**
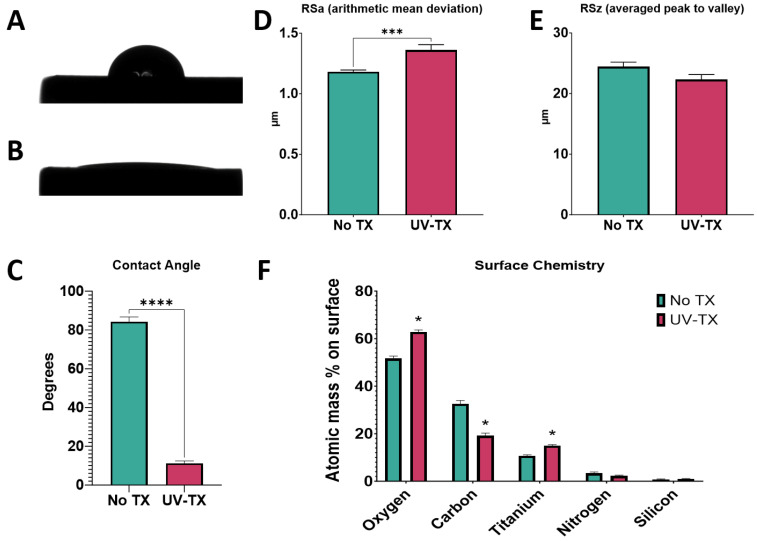
UV treatment effect on surface properties. Implant surface characterization shows increased wettability following treatment with UV–plasma-based cleaner. Sessile water droplet test of Ti6Al4V surface (**A**) and Ti6Al4V surface treated with UV–plasma cleaner (**B**). Contact angle measurements of water droplets for treated and untreated surfaces (**C**); measures were taken at 6 different locations on the implant surface. Optical profilometry measurements of surface micro-roughness (**D**) and peak-to-valley height (**E**). X-ray photoelectron spectroscopy to assess concentrations of elements present on the surface (**F**). Results are the means of 6 measurements taken at different points on 2 surfaces (*n* = 12) with bars showing SEM. Groups labeled with “*” are statistically significant compared to untreated Ti6Al4V using a Student’s unpaired *t*-test. (* = α < 0.05, *** = α < 0.0005, **** = α < 0.0001).

**Figure 2 biomimetics-09-00227-f002:**
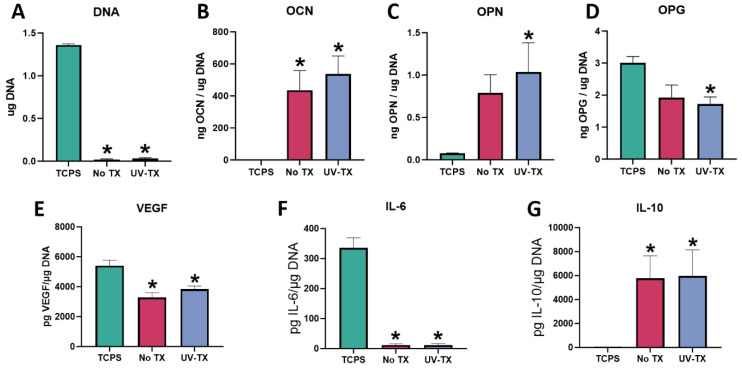
UV treatment effect on cell response. In vitro assessment of bMSCs cultured on UV–plasma-treated and untreated Ti6Al4V surfaces. Total DNA content measured at 7 days of culture (**A**). ELISA quantification of osteoblast maturation markers osteocalcin (**B**) and osteopontin (**C**), and paracrine signaling factors osteoprotegerin (**D**) and vascular endothelial growth factor (**E**) in response to UV–plasma-treated surfaces. Immunomodulatory cytokine production of IL-6 (**F**) and IL-10 (**G**). Groups are means of 6 cultures/variables, with errors bars representing SEM. Factor production in the conditioned media was normalized to total DNA and statistics were determined by ANOVA with Tukey post-test. Groups labeled with “*” are statistically significant compared to TCPS at *p*-value equal to or less than 0.05.

**Figure 3 biomimetics-09-00227-f003:**
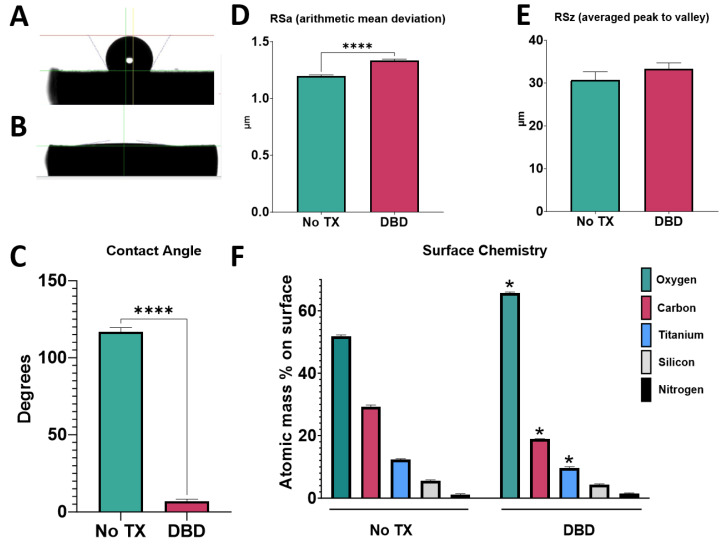
DBD treatment effect on surface properties. Implant surface characterization shows increased wettability following treatment with argon-based plasma cleaning method. Sessile water droplet test of Ti6Al4V surface (**A**) and Ti6Al4V surface treated with argon plasma cleaner (**B**). Contact angle measurements of water droplets for treated and untreated surfaces (**C**); measures were taken at 6 different locations on the implant surface. Optical profilometry measurements of surface micro-roughness (**D**) and peak-to-valley height (**E**). X-ray photoelectron spectroscopy to assess concentrations of elements present on the surface (**F**). Results are the means of 6 measurements taken at different points on 2 surfaces (*n* = 12) with bars showing SEM. Groups labeled with “*” are statistically significant compared to untreated Ti6Al4V using a Student’s unpaired *t*-test (* = α < 0.05, **** = α < 0.0001).

**Figure 4 biomimetics-09-00227-f004:**
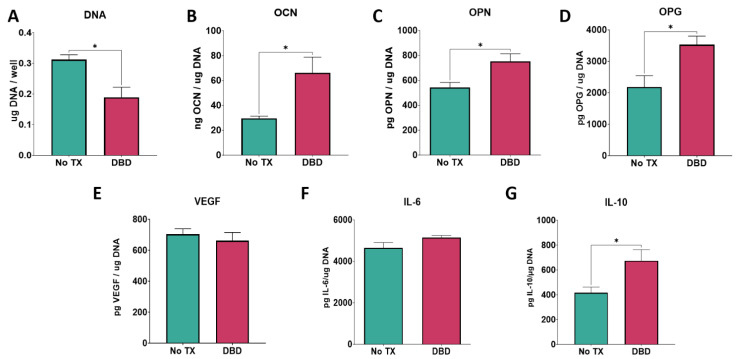
DBD treatment effect on cell response. In vitro assessment of bMSCs cultured on argon plasma-treated and untreated Ti6Al4V surfaces. Total DNA content measured at 7 days of culture (**A**). ELISA quantification of osteoblast maturation markers osteocalcin (**B**) and osteopontin (**C**), and paracrine signaling factors osteoprotegerin (**D**) and vascular endothelial growth factor (**E**) in response to argon plasma-treated surfaces. Immunomodulatory cytokine production of IL-6 (**F**) and IL-10 (**G**). Groups are the means of 6 independent cultures/variables, with error bars representing SEM. Factor production in the conditioned media was normalized to total DNA, and stats were determined using a Student’s unpaired *t*-test. Groups labeled with “*” are statistically significant compared to untreated Ti6Al4V at *p*-value equal to or less than 0.05.

**Figure 5 biomimetics-09-00227-f005:**
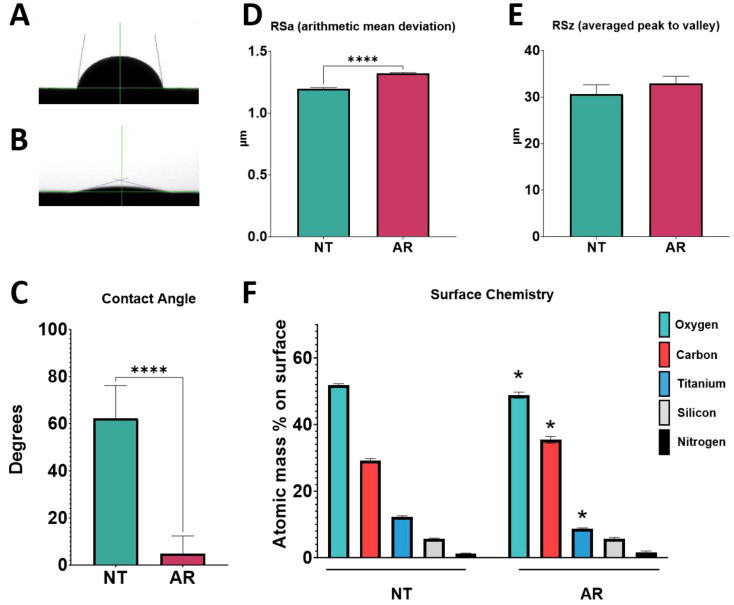
Argon treatment effect on surface properties. Implant surface characterization shows increased wettability following treatment with oxygen plasma-based cleaner under vacuum conditions. Sessile water droplet test of Ti6Al4V surface (**A**) and Ti6Al4V surface treated with UV–plasma cleaner (**B**). Contact angle measurements of water droplets for treated and untreated surfaces (**C**); measures were taken at 6 different locations on the implant surface. Optical profilometry measurements of surface micro-roughness (**D**) and peak-to-valley height (**E**). X-ray photoelectron spectroscopy to assess concentrations of elements present on the surface (**F**). Results are the means of 6 measurements taken at different points on 2 surfaces (*n* = 12), with bars showing SEM. Groups labeled with “*” are statistically significant compared to untreated Ti6Al4V using a Student’s unpaired *t*-test (* = α < 0.05, **** = α < 0.0001).

**Figure 6 biomimetics-09-00227-f006:**
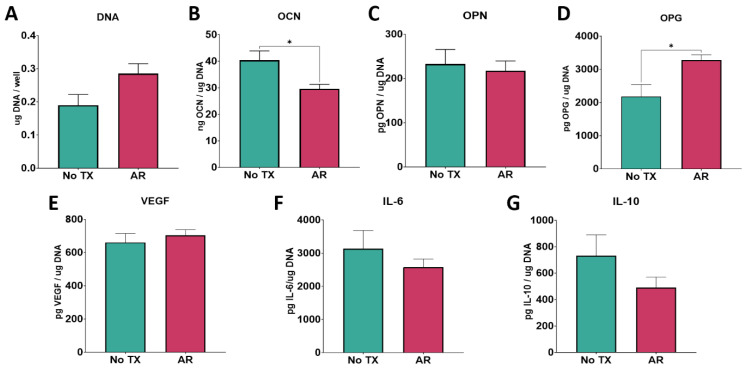
Argon treatment effect on cell response. In vitro assessment of bMSCs cultured on oxygen plasma under vacuum-treated and untreated Ti6Al4V surfaces. Total DNA content measured at 7 days of culture (**A**). ELISA quantification of osteoblast maturation markers osteocalcin (**B**) and osteopontin (**C**), and paracrine signaling factors osteoprotegerin (**D**) and vascular endothelial growth factor (**E**) in response to oxygen plasma vacuum-treated surfaces. Immunomodulatory cytokine production of IL-6 (**F**) and IL-10 (**G**). Groups are the means of 6 independent cultures/variables, with error bars representing SEM. Factor production in the conditioned media was normalized to total DNA, and stats were determined using a Student’s unpaired *t*-test. Groups labeled with “*” are statistically significant compared to untreated Ti6Al4V at *p*-value equal to or less than 0.05.

**Figure 7 biomimetics-09-00227-f007:**
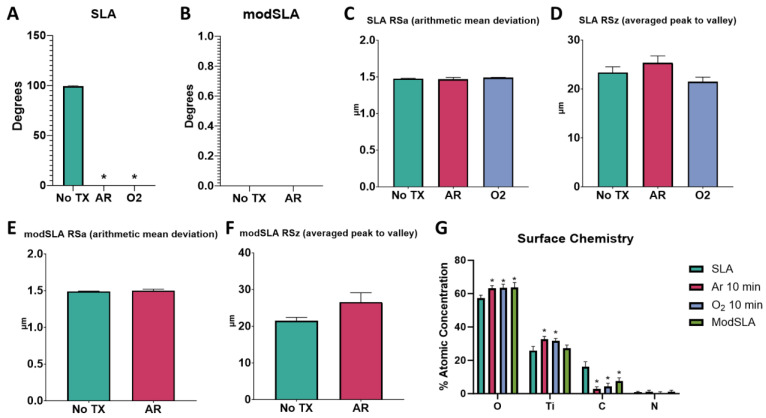
Argon and oxygen plasma treatment effect on surface properties of SLA surfaces. Surface characterization of SLA and modSLA surfaces that were treated with argon or oxygen plasma. Contact angle measurements of water droplets for treated and untreated SLA (**A**) and modSLA (**B**) surfaces; measures were taken at 6 different locations on the implant surface. Analysis of SLA surface micro-roughness (**C**) and peak-to-valley height (**D**) using optical profilometry. Optical profilometry measurements of surface micro-roughness (**E**) and peak-to-valley height (**F**) of modSLA-treated and untreated surfaces. X-ray photoelectron spectroscopy to assess concentrations of elements on untreated SLA and modSLA surfaces and plasma-treated SLA surfaces (**G**). Results are the means of 6 measurements taken at different points on 2 surfaces (*n* = 12), with bars showing SEM. Groups labeled with “*” are statistically significant compared to untreated SLA at *p*-value equal to or less than 0.05.

**Figure 8 biomimetics-09-00227-f008:**
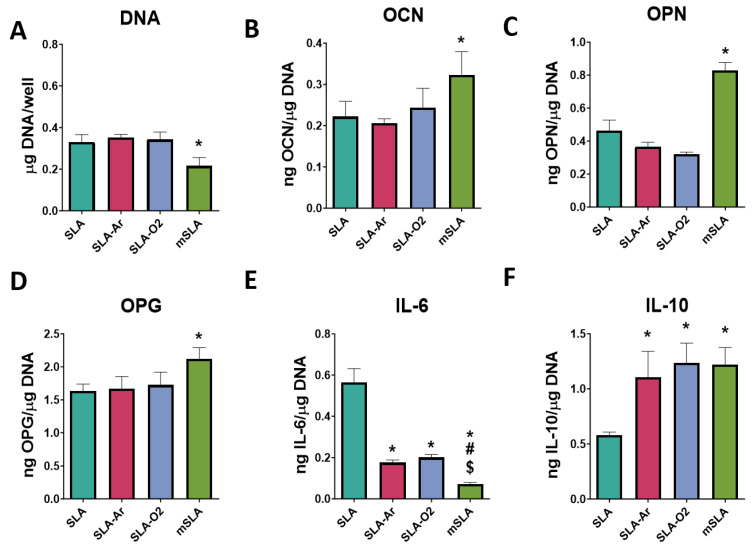
Argon and oxygen plasma treatment effect on cell response of SLA surfaces. In vitro assessment of bMSCs cultured on SLA surfaces treated with or without plasma and compared to modSLA. Total DNA content measured at 7 days of culture (**A**). ELISA quantification of osteoblast maturation markers osteocalcin (**B**) and osteopontin (**C**), paracrine signaling factor osteoprotegerin (**D**), and immunomodulatory cytokines Il-6 (**E**) and Il-10 (**F**) in response to SLA surfaces that were treated with either argon or oxygen plasma cleaner and compared to modSLA surfaces. Groups are the means of 6 independent cultures/variables with error bars representing SEM. Factor production in the conditioned media was normalized to total DNA, and stats were determined by ANOVA with Tukey post-test. Groups labeled with “*” are statistically significant compared to SLA at *p*-value equal to or less than 0.05. Groups labeled with a “#” are statistically significant compared to SLA-AR at *p*-value equal to or less than 0.05. Groups labeled with a “$” are statistically significant compared to SLA-O_2_ at *p*-value equal to or less than 0.05.

**Figure 9 biomimetics-09-00227-f009:**
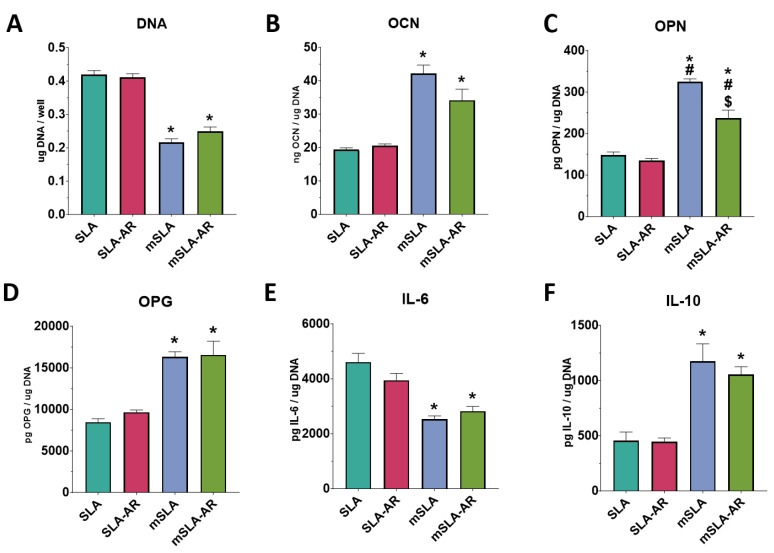
Argon plasma treatment effect on cell response of SLA and modSLA surfaces. In vitro assessment of bMSCs cultured on SLA and modSLA surfaces and treated with argon plasma. Total DNA content (**A**) and production of osteogenic markers osteocalcin (**B**), osteopontin (**C**), and osteoprotegerin (**D**) were measured. Production of cytokines Il-6 (**E**) and Il-10 (**F**) were measured. Groups are means of 6 independent cultures/variables, with error bars representing SEM. Factor production in the conditioned media was normalized to total DNA and stats were determined by ANOVA with Tukey post-test. Groups labeled with “*” are statistically significant compared to SLA at *p*-value equal to or less than 0.05. Groups labeled with a “#” are statistically significant compared to SLA AR at *p*-value equal to or less than 0.05. Groups labeled with a “$” are statistically significant compared to mSLA at *p*-value equal to or less than 0.05.

## Data Availability

The data generated in this study are available upon reasonable request from the corresponding author.

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
