# Peer review of "Different Methods to Modify the Hydrophilicity of Titanium Implants with Biomimetic Surface Topography to Induce Variable Responses in Bone Marrow Stromal Cells"

_biomimetics, 2024, doi:10.3390/biomimetics9040227_

Round 1

Reviewer 1 Report

Comments and Suggestions for Authors

The work doesn't fall in the aim and scope of the journal. Reviewers observed following parameters to be worked to enhance the quality of the manuscript.

·       Authors at the outset informed to go through the instructions of author for clarity such as research article should have 4000 words.

·       Graphical abstract is mandatory file which is missing in the manuscript.

·       Abstract underscores the content of work as the flow of description doesn’t seems will attract any reader. The description without any statistical data will connect the reader to context of work. Existing gaps must be highlighted like why this work is necessary? Abstract focus on some tangible numerical outcome value to justify the work carried out.

·       Title looks like a generic statement in the description. It doesn’t enlighten about the holistic amount of work carried out. Refine and make it concise with crisp keywords.

·       Authors are expected to explain clearly how this work falls within the purview of journal aim and scope. As with the abstract it better suites to materials or metals based journal. Justify

·       The manuscript has 25 references out of which mere 04 articles were from 2021 and onwards. This shows the latest technology upgradations have been missed out in the list.

·       Literature survey in the introduction seems to be limited. Recent articles have to be considered for comparative study.

·       Keywords seems to be too many restrict it to 4 to 6

·       There are no details about process map or road map for the entire work in methodology? Do provide an additional figure comprising the same. Even the journal requirement is that you need to have a graphical abstract.

·       There are many typo errors or grammatical corrections do check the manuscript in detail.

·       SI units should be used throughout.

·       Plagiarism and Self-citation need to be checked.

·       For all the experiments standard deviation need to be mentioned with tolerance limit.

·       There were in total 8 different test conducted but not even a single citation being added with earlier work of predecessor. Which is not at all acceptable.

·       Reviewers still feel that the entire work doesn’t fall within the aim and scope of journal.

·       Results and discussions are acceptable only with any correlation between the current used material and existing models in public domain. Such as predecessors work on the typical different kind of materials.

·       There is no comparative study with analytical/experimental/simulation for validation of the extracted results. Validation analysis is missing the comparative study not even a single citation is considered inside this. The discussion section lags in explanation with respect to the work carried out.  there are limited number of citations in discussion section to compare the work with existing materials.

·       Conclusion looks to be generic need to compile the outcomes and state based on the tests conducted and convey how best this can fit in the current context for any application. In conclusion section, values must be displayed with explanation. It's better to mention the salient features of the entire work in terms of bullet points with current context.

Comments on the Quality of English Language

There are few corrections as per English language. Please look into the sentence formations.

Reviewer 2 Report

Comments and Suggestions for Authors

Dear authors.

Many thanks for your work. Comparison of different plasma treatment systems is very important topic. Unfortunately, there is a lot of not fully correct information presented about the devices and treatment conditions, so there is very difficult to obtain correct information from the results. The most important is my last comment that summarizes what is necessary to complete before publication. The current form of paper is non-acceptable because of high leak of important information.

Term “plasma treatment method” should be narrowed because there are many different plasma treatment techniques

Abstract – what “atmospheric argon or oxygen” does mean? Simply presence of gas or some electrical discharge in them? Also for DBD, the working gas should be pointed out.

Section 2.1 at sand blasting “(250-500 m corundum)” unit is not described properly. If dimension is in microns, I’m thinking that this produces rather rough surface, may be more tiny grains will produce better surface structure, but I have no experimental evidence for this, unfortunately.

Section 2.2 “Plasma treatment was initiated using a radio frequency of 13.56 MHz with maximum power output of -27.46 dBM and UV exposure was set to 172 nm wavelength. Treatment duration was 10 seconds.” – what was gas and pressure? If implants were inside the plasma, not only radiation (proper term will be VUV here, because 172 nm is not propagating through air) can take place. What was pressure and working gas? Based on radiation, is seems be some excimer mixture. Surely, there is not only one wavelength because radiation is much broader. This must be clarified. Probably, 172 nm is some maximum of broad peak. Power should be given in Watts. Last point of this section: Implants ware placed at electrode (what one, with/without bias) or there in the space at the floating potential?

Section 2.3 – What was working gas and applied power or power density? The use of polypropylene as dielectric barrier seems be surprising because of its fast degradation by plasma generated UV and reactive species. Did not contamine the polypropylene degradation products the implant surface? And what was discharge character – homogeneous or streamer like.

Section 2.4 – specify following points: Electrode configuration, operation pressure, current regime (DC or AC, pulsed or continuous), samples were part of electrode (what one), etc.

Section 2.5.1 – “56 A ion current” symbol micro is not shown. The same is repeated in whole text.

Section 2.5.4 – XPS had to be used also in high resolution mode to obtain information about chemical bonds at the samples surface. I’m thinking that after plasma treatment should be also bonded some nitrogen or oxygen (or OH) that can significantly modify the surface properties.

Fig. 1 – symbol TX can be omitted. Simply use non-treated/treated. Is any change of titanium and oxygen peaks ratio? Generally, surface seems be from titanium dioxide. Why there is no presence of aluminium and vanadium, that are in the bulk material of implants? And how and how long were samples stored between plasma treatment and XPS analysis? Non-correct storage can totally change the results. Also, ware cell cultures applied immediately or with some delay on the treated samples?

Section 3.1.2 What is conclusion from the obtained experimental data? Has treatment positive effect or neutral or even negative in some markers?

Section 3.2. title – “vacuum DBD is not fully correct term because 750 Pa is not vacuum, this is moderate pressure

Why native material WCA is so different in section 3.1 and 3.2?

Fig. 3 – there is still high content of carbon at the sample surface. Is this carbon bonded to surface or it is simply non-bounded contamination? This can be distinguished based on high resolution XPS of carbon, titanium and oxygen peaks. Additionally, the high oxygen content did not reflect TiO2 stoichiometry, co presence of CO groups is probable. Where is the source of silicon? It is not in the alloy. On the other hand, there are peaks related to aluminium and vanadium?

What is conclusion of section 3.2.2.?

Section 3.3 – the same comments as for the section 3.2. Additionally, why carbon content is increased by the plasma treatment?

Fig. 7g – XPS is not measuring molecular oxygen but atomic oxygen.

Sections 3.4.2 and 3.4.3 needs some discussion why is observed what was observed.

It is true that cell response to surfaces is related more to their chemical composition than physical changes of surfaces. But simple elementary composition of surface is non-sufficient information about chemical composition. This must be measured correctly using high resolution of selected XPS peaks. Without this, all discussion is very speculative. Also, with respect to the cells or proteins adhesion to surface, simple water contact angle is non-sufficient because this gets no information about polar/non-polar component of surface energy. So, more liquids and another model for surface energy calculation must be applied. This will allow also calculation of acid and base components of surface energy. Additionally, the cell grow medium is rather complex and thus its contact angle can also greatly differ from the WCA.

Reviewer 3 Report

Comments and Suggestions for Authors

Comments to the Author
The authors of the manuscript “biomimetics-2831481-peer-review-v1” presented an interesting investigating of the biological responses to Hydrophilic Titanium Implant Surfaces vary with plasma treatment methods.  However, before the manuscript is to be published, authors are required to respond to the following inquiries.  

-          The abstract needs to be re-written in more quantitative analysis. The plasma properties varied depending on the power scheme and discharge gas type. So, please be specific, which plasma type gave the best performance.

-          The subtitle “UV-Plasma Treatment” is ambiguous for me. UV is one of the plasma components, hence it would better to differentiate between the UV effect and plasma effects. UV is photons with high energy, while plasma is composed of high energetic electrons, ions, UV, and radicals. The Plasma effects on substrates differ that the UV effect.  Please more explanation on the RF plasma source specified on subtitle 2.2.

-          Discharge power unite, page 3, line 147…. please convert the unit from dBM to W or KW, and elsewhere in the manuscript.

-          Remove the word cleaning from the subtitle “Dielectric Barrier Discharge Plasma.”

-          Authors are recommended to include the SEM images for the samples before and after acidic /plasma treatments.

-          Authors are recommended to include the survey XP spectra of the samples before and after the plasma treatments.

-          Page 7, subtitle “3.2”, remove the word “vacuum.”

Reviewer 4 Report

Comments and Suggestions for Authors

In this paper, the authors compare the effects of plasma treatment on surfaces made of Ti grade 2 and Ti6Al4V and investigate its effect on surface properties. Considering the current state of knowledge, the topics of the article are up-to-date and the systematisation of the knowledge that has been included in the article is much needed. The article bond the material science conventional methods with biological response analysis which is of great advantage of this research. I am happy to recommend this article for publication, however, I have some minor comments which I include below:

- line 138 the micrometer symbol is not visible in .pdf

- supplementary materials: the images presented in the supplementary has very small indication of scale (supp. figure 3) and unreadable scale in supp. figure 4. authors should make it possible to read, especially the mag and scale bars.

- In the article there is a lack of conclusion section. I propose to add conlusions and sum up the results in 3-5 main founding and observation that generalise the whole research

- The overall count of citation is 25, nevertheless the researches which dissussed about simillar investigation is quite heavy in number. I propose to check and dissuss on more quantiti of the articles.

- In the article from my point of view there should be the image of the representation of the material as it is disscussed. I am affraid that moving every microscopic image to the supplementary materials is not the best option for the reader

Reviewer 5 Report

Comments and Suggestions for Authors

The manuscript investigates the plasma treatment of titanium implant, the topic is interesting.

1.        One of the main concern raises around the novelty of this this method is well-known globally. Therefore, I suggest authors to address the novelty and originality of this topic in introduction.

2.        Generally, atmospheric pressure plasmas are suitable for treatment of organic materials as the weaker bonds can be easily break. How and why, they choose DBD for treatment for metals. Why treatment at 5 Torr has been selected? It would be better to address this in the body of the manuscript.

3.        It is known that distance of which ions can go has inverse relation with pressure, it means that the thickness of the treated surface should be very thin. Do authors consider performing any characterization on cross-section of the samples?

4.        Is this treatment controllable and how the results reproducible?

5.        The entire manuscript there is lack of putting the quantities in evidence and I missed mentioning them in the text.

6.        Regarding the results, it would give deeper information if the XPS high-resolution spectra have been presented.

7.        How authors can explain the morphology and roughness: since SEM showed almost no effects and roughness showed some changes (though there is not quantity mentioned in the text).

Round 2

Reviewer 1 Report

Comments and Suggestions for Authors

Authors have successfully modified the comments and recommend the authors to pass the draft for English expert proofread and then it can be considered for acceptance.

Comments on the Quality of English Language

English proofread is required for fine tuning the grammatical errors.

Reviewer 2 Report

Comments and Suggestions for Authors

Dear authors!

Many thanks for the substantial changes in the manuscript. The current forms seems be acceptable.

Comments on the Quality of English Language

There are some minor typo problems, besides them all is acceptable.

Reviewer 3 Report

Comments and Suggestions for Authors

I have reviewed the revised Biomimetics-283148 manuscript. The authors did a real effort to improve the manuscript.  They have answered my inquiries and I accept them. Therefore, I recommend the last revised version for publication in Biomimetics 

Reviewer 5 Report

Comments and Suggestions for Authors

The authors addressed the main concerns, therefore,  the manuscript can be considered for publication. 
